# Association between mild cognitive impairment and lumbar degenerative disease in a Japanese community: A cross-sectional study

Kazushige Koyama[1], Kanichiro Wada[1]*, Gentaro Kumagai[1], Hitoshi Kudo[1], Sunao Tanaka[1], Toru Asari[1], Songee Jung[2], Masataka Ando[3], Yasuyuki Ishibashi[1]

1 Department of Orthopaedic Surgery, Graduate School of Medicine, Hirosaki University, Hirosaki, Aomori, Japan, 2 Department of Digital Nutrition and Health Sciences, Graduate School of Medicine, Hirosaki University, Hirosaki, Aomori, Japan, 3 Department of Diet and Health Sciences, Graduate School of Medicine, Hirosaki University, Hirosaki, Aomori, Japan

* wadak39@hirosaki-u.ac.jp

**Data Availability Statement:** This study was conducted at one small town of Iwaki Prefecture in Japan. The anonymized data set used for this study

## Abstract

Lumbar degenerative disease and dementia are increasing in super-aging societies and are both related to physical dysfunction and pain. However, the relationship between these diseases remains unclear. This cross-sectional study aimed to investigate the comorbidity rates of lumbar spinal canal stenosis (LSS) and mild cognitive impairment (MCI) and clarify the association between LSS presence, lumbar symptoms, and quality of life (QOL) related to low back pain and cognitive impairment in the Japanese population. We enrolled 336 participants (men 124; women 212; mean age 72.2 years) from a medical checkup program. LSS was diagnosed using a self-administered questionnaire, and lumbar symptoms were evaluated using the visual analog scale (low back pain, and pain and numbness of the lower limb). QOL related to low back pain was evaluated using the Japanese Orthopedic Association Back-Pain Evaluation Questionnaire (JOABPEQ: pain, and lumbar, and gait function). Radiological lumbar degeneration was classified using Kellgren-Lawrence grading and lateral radiographs of the lumbar spine. Cognitive function was measured using the Mini Mental State Examination (MMSE), and MCI was defined by a summary score of MMSE ≤27. Logistic and multiple linear regression analyses were performed to analyze the association between MCI, summary score of MMSE, and lumbar degenerative disease. The comorbidity rate of MCI and LSS was 2.1%, and the rate of MCI was 41% in participants with LSS. Lumbar function in JOABPEQ was associated with MCI. The presence of LSS and lumbar function in JOABPEQ were associated with MMSE. Over one-third of the people with LSS had MCI. The presence of LSS and deterioration of QOL due to low back pain were related to cognitive impairment. We recommend evaluating cognitive function for patients with LSS because the rate of MCI was high in LSS participants.

still contains private information, i.e. combination of sex, age, BMI and preferences including alcohol and smoking has a possibility to lead to identification of individuals participating in the survey. Data cannot be shared publicly because of the ethical concerns. Data are available from the Hirosaki University COI Program Institutional Data Access / Ethics Committee (contact via e-mail: coi@hirosaki-u.ac.jp) for researchers who meet the criteria for access to the data. Researchers need to be approved by research ethics review board at the organization of their affiliation.

**Funding:** This study was supported in part by a Grant-in-Aid from the Ministry of Education, Culture, Sports, Science and Technology of Japan (No. 18200044), the Japanese Society for the Promotion of Science (No.21500676), a Health and Labour Sciences Research Grant by JOA-Subsidized Science Project Research from the Japanese Orthopaedic Association (No. 2015-02), and the Iwaki Health Promotion Project Center of Innovation Program (No. JPMJCE1302).

**Competing interests:** The authors have declared that no competing interests exist.

## Introduction

Lumbar degenerative disease, such as lumbar spinal canal stenosis (LSS), degenerative spondylolisthesis, and spondylosis deformity, is more prevalent among middle-aged and elderly individuals. Lumbar degenerative disease causes low back pain, intermittent claudication, bladder and rectal disturbance, numbness and lower limb pain, and deterioration in muscle power of the lower limbs as well as standing and walking abilities [1, 2]. The number of people whose quality of life (QOL) is impaired by lumbar degenerative disease is expected to increase in a super-aging society; the proportion of older adults whose age is ≥65 years is over 21%.

Mild cognitive impairment (MCI) is a pre-stage of dementia, and people with MCI are at high-risk for developing dementia [3]. However, MCI can be improved if risk factors of cognitive impairment are reduced [4, 5]. Thus, reducing the risk of cognitive impairment is important to prevent the development of dementia. Several reports have shown that cognitive impairment is associated with pain and physical dysfunctions [6–8], which are the typical symptoms of musculoskeletal diseases. Cognitive function certainly deteriorates with age, thereby increasing the prevalence of MCI in the future.

Previous studies investigating the association between cognitive function and musculoskeletal diseases have reported that MCI was related to the occurrence of knee osteoarthritis [9] and that delayed union of osteoporotic vertebral fracture decreased cognitive function [10]. However, the association between lumbar degenerative disease and cognitive function remains unclear, although both lumbar degenerative disease and cognitive impairment are related to pain and physical dysfunction. Hence, clarifying this relationship may be important for improving activities of daily living in older adults. This study aimed to investigate the comorbidity rates of MCI and LSS by age and sex, and to clarify the relationship between MCI/ MMSE score and the factors related to lumbar degenerative disease—LSS presence, lumbar symptoms, QOL score depending on low back pain, and radiographical lumbar degeneration in a community-dwelling Japanese population.

## Materials and methods

### Participants and study design

Our analysis was based on data collected from a medical checkup program in a small city in northern Japan, the Iwaki Health Promotion Project (UMIN000040459), in 2016. In brief, this program was initiated in 2005, and it was conducted over a 10-year period. In addition to orthopedic surgeons, gynecologists, urologists, endocrinologists, cardiologists, gastroenterologists, neurologists, and otolaryngologists were involved in this project. As one aspect of the multiple-focused check, we collected questionnaires and radiographical images related to musculoskeletal disorder.

For this medical checkup program, 1,149 participants were enrolled (20 to 93 years old). Participants were asked to complete self-administered questionnaires assessing their daily habits, medical histories, and lumbar degenerative disease. They also underwent a cognitive screening test and a lateral lumbar radiograph in the neutral position. For this study, we included only participants aged ≥65 years and excluded those who did not answer the questionnaire entirely, did not undergo the cognitive screening test and radiographical examination, or had medical histories of lumbar spine surgeries. Of the 354 participants aged ≥65 years, 18 were excluded. Of those excluded, 16 (6 men and 10 women) did not answer the questionnaire entirely, 2 (1 man and 1 woman) did not undergo radiographical examination, and 0 had previous lumbar surgery. Finally, 336 people (124 men and 212 women, mean age, 72.2 years) were included in this study. All participants were thoroughly explained that the

data collected would be analyzed and they gave written informed consent for participation. This cross-sectional study was approved by the ethics committee of the Hirosaki University Graduate School of Medicine.

## Diagnosis of LSS

LSS was diagnosed by using a simple single sheet, self-administered, self-reported history questionnaire. It consisted of 15 questions associated with buttock and lower limb numbness or pain, and intermitted-claudication. Each question was scored from -1 to 5 points. We totaled the points of all questions and diagnosed LSS+ when the score was 13 points or greater. The sensitivity and specificity were 92.7% and 84.7% for diagnosing LSS in patients with lower limb symptoms due to LSS, arteriosclerosis obliterans, diabetes, or peripheral nerve diseases [11].

## Evaluation of cognitive function

Cognitive function was measured using the Mini Mental State Examination (MMSE) [12]. This is a 30-item cognitive screening test that measures orientation, registration, short-term memory, attention and concentration, and language and construction capacity. The full score of MMSE is 30 points, with 0 defining the worst cognitive function. We defined the participant as having MCI when the MMSE score was 27 points or less, as this cut-off value had good sensitivity (66.3%) and specificity (72.9%) for diagnosing MCI in a previous report [13]. We classified participants whose MMSE score was 27 points or less as the MCI+ group, and participants whose MMSE score was over 27 points as the MCI- group.

## Assessment of lumbar symptom and QOL associated with low back pain

The severity of the lumbar symptom was evaluated with the visual analog scale (VAS) for the most severe low back pain, lower limb pain, and numbness during the past three months. The most severe score is 100 mm, and 0 mm means no symptoms.

Deterioration of QOL due to low back pain was evaluated using the Japanese Orthopedics Association Back Pain Evaluation Questionnaire (JOABPEQ) [14]. There have been several reports that the JOABPEQ is useful for evaluating QOL in patients with lumbar disc herniation, LSS, and other lumbar diseases [15–17]. It consists of five domains: pain, gait function, lumbar function, social life, and mentality, and each domain is scored out of 100 points. Three of the five domains: pain, gait function, and lumbar function, were used for analysis in this study. Research assistants supported the participants who could not answer these questionnaires by themselves.

**Measurement of lumbar degeneration on radiography.**   Lateral radiographs of the lumbar spine were taken, with the participants standing naturally, with their forearms crossed and hands on the chest. Radiographs were evaluated by a single orthopedic surgeon (KK) using Kellgren-Lawrence grading [18] in each intervertebral level (L1/2, L2/3, L3/4, L4/5, and L5/ S1). To determine the severity of lumbar degeneration, the values of Kellgren-Lawrence grading were summed from L1/2 to L5/S. According to this summed value, 0 corresponded to a normal lumbar spine and 20 expressed the most severely degenerative lumbar spine [19].

## Medical histories and daily habits

All participants provided data related to their medical histories and daily habits, which were previously reported to be related to cognitive impairment [20–22]. The Center for Epidemiologic Studies Depression Scale was used to assess the prevalence of depressive symptoms. This

scale is a short-self report scale designed to measure depressive symptomatology in the general population. We defined depression as 16 points or greater on this scale [23]. Medical histories related to diabetes and hypertension were collected. We collected data pertaining to their duration of education (6 to 20 years), daily smoking habits (0, ex-smoker or never smoked; 1, current smoker), alcohol consumption (0, ex-drinker or never drank or social drinker; 1, habitual drinker), and exercise (0, no habit of exercising; 1, exercises over 2 times during one week).

## Statistical analyses

All investigated factors were compared by sex using the Mann–Whitney U test. To compare characteristics between MCI+ and MCI- groups, we used the Mann–Whitney U test, with the presence of MCI as the dependent variable and age, sex, body mass index, VAS, JOABPEQ, summed Kellgren-Lawrence grading, education periods, diabetes, hypertension, depression, smoking, alcohol consumption, and exercise as independent variables. To analyze the correlation between MCI and lumbar degenerative disease parameters, logistic regression analysis was performed with MCI as the dependent variable and the presence of LSS, VAS, JOABPEQ, and summed Kellgren-Lawrence grading as independent variables. For adjusting parameters related to cognitive function, age, sex, education periods, diabetes, hypertension, depression, smoking, alcohol consumption, and exercise were also included as independent variables. To analyze the correlation between MMSE and lumbar degenerative parameters, multiple linear regression analysis was performed with MMSE as the dependent variable and the presence of LSS, VAS, JOABPEQ, and summed Kellgren-Lawrence grading as independent variables. All statistical tests were performed using SPSS ver. 22.0 (SPSS Inc., Chicago, IL, USA), and statistical significance was set at 0.05.

## Results

### Prevalence of MCI and LSS

Of the 336 participants, 72 (21.4%) had MCI, and the number of men in the MCI group was significantly larger compared with women (Table 1). Seventeen individuals (5.1%) were diagnosed with LSS, and there was no significant difference between sexes. The comorbidity rate of MCI and LSS was 2.1% (7/336) among all the participants, 2.4% in men, and 1.9% in women. The rate of MCI was 41% (7/17) in participants with LSS, and the rate of LSS was 9.7% (7/72) in participants with MCI.

### Comparison of characteristics between MCI+ and MCI- groups

We investigated the differences in parameters between the MCI+ group and the MCI- group using the Mann–Whitney U test. Age, number of men, and the prevalence of LSS and summed Kellgren-Lawrence grading were significantly higher in the MCI+ group than in the MCI- group. Three domains of JOABPEQ were lower in the MCI+ than in the MCI- group (Table 2).

### The association between the presence of LSS, lumbar symptom, and QOL related to low back pain and MCI or MMSE

Logistic regression analysis with MCI as the dependent variable established an association between the parameters of lumbar degenerative disease (presence of LSS, lumbar symptom, and QOL score related to low back pain) and MCI. Only the lumbar function in JOABPEQ (Odds ratio 0.985, 95% confidence interval (CI) 0.972 to 0.997) was significantly associated with MCI (Table 3). The associations between the parameters of lumbar degenerative disease

**Table 1. Differences in MCI and LSS prevalence, JOABPEQ, VAS, medical histories, and daily habits with sex.**

| | Total (336) | | Men (124) | | Women (212) | | p |
|---|---|---|---|---|---|---|---|
| | mean or n | SD or % | mean or n | SD or % | mean or n | SD or % | |
| Age (year) | 72.2 | 6.0 | 72.3 | 6.4 | 72.2 | 5.8 | 0.99 |
| Body mass index (kg/m$^2$) | 23.3 | 3.1 | 23.4 | 2.9 | 23.3 | 3.2 | 0.30 |
| LSS+ | 17 | 5.1 | 4 | 3.2 | 13 | 6.1 | 0.24 |
| JOABPEQ | | | | | | | |
| *pain* | 82.0 | 28.3 | 82.9 | 29.1 | 81.4 | 27.9 | 0.39 |
| *lumbar function* | 84.1 | 20.9 | 86.2 | 19.0 | 82.9 | 21.9 | 0.16 |
| *gait function* | 82.7 | 25.0 | 87.7 | 20.6 | 79.7 | 26.8 | 0.001 |
| VAS | | | | | | | |
| *low back pain* | 22.3 | 24.7 | 19.9 | 23.6 | 23.7 | 25.3 | 0.16 |
| *pain of lower limbs* | 7.9 | 17.9 | 7.8 | 18.9 | 7.9 | 17.3 | 0.99 |
| *numbness of lower limbs* | 7.8 | 18.7 | 7.2 | 18.6 | 8.1 | 18.7 | 0.82 |
| Summed Kellgren-Lawrence grading | 6.9 | 4.1 | 8.3 | 4.1 | 6.1 | 3.9 | <0.001 |
| MMSE | 28.5 | 2.0 | 28.2 | 2.2 | 28.6 | 1.8 | 0.15 |
| MCI+ | 72 | 21.4 | 35 | 28.2 | 37 | 17.5 | 0.020 |
| Education periods (year) | 10.9 | 2.1 | 11.3 | 2.3 | 10.7 | 2.0 | 0.022 |
| Diabetes | 37 | 11.0 | 21 | 16.9 | 16 | 7.5 | 0.008 |
| Hypertension | 187 | 55.7 | 77 | 62.1 | 110 | 51.9 | 0.070 |
| Depression | 71 | 21.1 | 35 | 28.2 | 36 | 17.0 | 0.020 |
| Smoking | 18 | 5.4 | 14 | 11.3 | 4 | 1.9 | <0.001 |
| Alcohol | 123 | 36.6 | 87 | 70.2 | 36 | 17.0 | <0.001 |
| Exercise | 73 | 21.7 | 33 | 26.6 | 40 | 18.9 | 0.10 |
| Comorbidity rate of LSS and MCI | 7 | 2.1 | 3 | 2.4 | 4 | 1.9 | 0.74 |

Data are presented as mean and SD or n and %. Comparisons between sexes were performed using the Mann–Whitney U test. SD, standard deviation; n, number of participants; MCI, mild cognitive impairment; LSS, lumbar spinal canal stenosis; JOABPEQ, Japan Orthopedics Association Back Pain Evaluation Questionnaire; VAS, visual analog scale; MMSE, Mini Mental State Examination; MCI, mild cognitive impairment.

and MMSE were shown using multiple linear regression analysis with MMSE as the dependent variable. The presence of LSS (standardized regression coefficient (β) = -0.120, 95% CI -1.972 to -0.169) and lumbar function in JOABPEQ (lumbar function: β = 0.191, 95% CI 0.008 to 0.027) were significantly related to MMSE (Table 4).

## Discussion

The prevalence of MCI and LSS in general populations has been reported previously. The prevalence of MCI varied from 3.0 to 42% and had a strong association with aging [24, 25]. The prevalence of MCI was 21.4% in this study and in line with other published researches. The range of prevalence of MCI, however, varied greatly between different population studies (including the present study)—one reason being that the diagnostic tools used for MCI were different in various studies, such as Clinical Dementia Rating, MMSE, Montreal Cognitive Assessment, Psychogeriatric Assessment Scale, or Wechsler Memory Scale-Revised. We defined MCI by a summary score of MMSE ≤ 27, in this study. MMSE is a popular and useful screening tool for evaluating cognitive function, and the cut-off value used in the present study had good sensitivity (66.3%) and specificity (72.9%) for diagnosing MCI [13]. The prevalence of LSS was 10.1% in men and 8.9% in women and was associated with age in the Japanese population study [26]. In that study, LSS was diagnosed using lumbar magnetic resonance image,

**Table 2. Differences in demographic characteristics between groups with and without MCI.**

| | MCI+ (72) | | MCI- (264) | | P |
|---|---|---|---|---|---|
| | mean or n | SD or % | mean or n | SD or % | |
| Age (year) | 75.3 | 6.9 | 71.4 | 5.5 | <0.001 |
| Men | 35 | 48.6 | 89 | 33.7 | 0.020 |
| Body mass index (kg/m$^2$) | 23.1 | 3.3 | 23.4 | 3.0 | 0.66 |
| LSS+ | 7 | 9.7 | 10 | 3.8 | 0.042 |
| JOABPEQ | | | | | |
| Pain | 75.8 | 32.4 | 83.7 | 26.9 | 0.045 |
| lumbar function | 77.5 | 26.0 | 85.9 | 19.0 | 0.012 |
| gait function | 76.0 | 27.3 | 84.5 | 24.1 | 0.004 |
| VAS | | | | | |
| low back pain | 25.1 | 26.9 | 21.5 | 24.1 | 0.49 |
| pain of lower limbs | 12.0 | 23.6 | 6.7 | 15.8 | 0.10 |
| numbness of lower limbs | 11.8 | 26.0 | 6.7 | 16.0 | 0.38 |
| Summed Kellgren-Lawrence grading | 8.0 | 4.1 | 6.6 | 4.1 | 0.006 |
| MMSE | 25.4 | 1.8 | 29.3 | 0.8 | <0.001 |
| Education periods (year) | 10.2 | 2.0 | 11.1 | 2.1 | <0.001 |
| Diabetes | 7 | 9.7 | 30 | 11.4 | 0.69 |
| Hypertension | 46 | 63.9 | 141 | 53.4 | 0.11 |
| Depression | 16 | 22.2 | 55 | 20.8 | 0.76 |
| Smoking | 3 | 4.2 | 15 | 5.7 | 0.61 |
| Alcohol | 27 | 37.5 | 96 | 36.4 | 0.86 |
| Exercise | 14 | 19.4 | 59 | 22.3 | 0.60 |

Data are presented as mean and SD or as n and %. Comparisons between groups with and without MCI were performed using the Mann–Whitney U test. SD, standard deviation; n, number of participants; MCI, mild cognitive impairment; LSS, lumbar spinal canal stenosis; JOABPEQ, Japan Orthopedics Association Back Pain Evaluation Questionnaire; VAS, visual analog scale; MMSE, Mini Mental State Examination; MCI, mild cognitive impairment.

**Table 3. The impact of parameters related to lumbar degenerative disease on MCI.**

| | Odds ratio | | 95% CI | | P |
|---|---|---|---|---|---|
| LSS+ | | | | | 0.15 |
| JOABPEQ | | | | | |
| pain | | | | | 0.13 |
| lumbar function | 0.985 | 0.972 | – | 0.997 | 0.017 |
| gait function | | | | | 0.18 |
| VAS | | | | | |
| low back pain | | | | | 0.79 |
| pain of lower limbs | | | | | 0.061 |
| numbness of lower limbs | | | | | 0.10 |
| Summed Kellgren-Lawrence grading | | | | | 0.69 |

Logistic regression analysis was performed with MCI as the dependent variable. Parameters related to lumbar degenerative disease were independent variables. For adjusting for age, sex, education periods, diabetes, hypertension, depression, smoking, alcohol consumption, and exercise were included as independent variables. MCI, mild cognitive impairment; LSS, lumbar spinal canal stenosis; JOABPEQ, Japan Orthopedics Association Back Pain Evaluation Questionnaires; VAS, visual analog scale; CI, confidence interval.

**Table 4. The impact of parameters related to lumbar degenerative disease on MMSE.**

|  | β |  | 95% CI |  | P |
|---|---|---|---|---|---|
| LSS+ | -0.120 | -1.972 | – | -0.169 | 0.020 |
| JOABPEQ |  |  |  |  |  |
| *pain* |  |  |  |  | 0.073 |
| *lumbar function* | 0.191 | 0.008 | – | 0.027 | <0.001 |
| *gait function* |  |  |  |  | 0.20 |
| VAS |  |  |  |  |  |
| *low back pain* |  |  |  |  | 0.57 |
| *pain of lower limbs* |  |  |  |  | 0.13 |
| *numbness of lower limbs* |  |  |  |  | 0.30 |
| Summed Kellgren-Lawrence grading |  |  |  |  | 0.76 |

Multiple linear regression analysis was performed with MMSE as the dependent variable, and parameters related to lumbar degenerative disease as independent variables. For adjusting for age, sex, education periods, diabetes, hypertension, depression, smoking, alcohol consumption, and exercise were included as independent variables. MMSE, Mini Mental State Examination; LSS, lumbar spinal canal stenosis; JOABPEQ, Japan Orthopedics Association Back Pain Evaluation Questionnaires; VAS, visual analog scale; CI, confidence interval.

medical histories, and physical tests performed by an orthopedic surgeon. The current study showed that the prevalence of LSS was 5.1% and was lower in our study than that in a previous study because the diagnostic tools used for LSS were different; we used a simple single sheet, self-administered, self-reported history questionnaire for diagnosing LSS. The diagnostic tool for LSS we used had good sensitivity (92.7%) and specificity (84.7%) for diagnosing LSS [11]. Thus, the diagnosing tools for MCI and LSS used in this study did not require considerable time and effort and were useful as screening tools, and the prevalence of MCI and LSS in this study were reliable enough. The comorbidity rate of frailty and MCI was 2.7% in a Japanese population [27]; however, to the best of our knowledge, there have been no reports on the comorbidity rates of MCI and LSS to date either in patients or the general population. The comorbidity rates of MCI and LSS were 2.1%, the rate of MCI was 41% in participants with LSS, and the rate of LSS was 9.7% in participants with MCI in this study. The risk factors related to LSS presence were age, diabetes, or low ankle-brachial index values [28, 29]. It was suggested that the prevalence of MCI in the participants with LSS and the prevalence of LSS in the participants with MCI were high because some risk factors related to cognitive impairment are similar to the risk factors related to LSS presence [20–22]. Because over 40% of patients with LSS might have cognitive impairment, we should bear in mind the effect of cognitive impairment on LSS symptoms when we treat patients with LSS. This study was held as a part of a medical checkup program, and almost all participants were interested in health. Thus, the comorbidity rates of MCI and LSS might be higher in people who are not interested in health and who cannot go outside because of deteriorating mobility; they were enrolled as participants in a similar study. The percentage of elderly persons in Japan will highly increase in the future, suggesting that the comorbidity rates of MCI and LSS will correspondingly increase further.

Several reports have shown the links between musculoskeletal diseases and cognitive function. Baseline MMSE summary score and the prevalence of MCI were significantly associated with the incidence of knee osteoarthritis [9]. Delayed union of vertebral fractures decreased MMSE summary score in a longitudinal study [10]. However, to the best of our knowledge, there has been no report on lumbar degenerative disease and cognitive impairment. This study was the first report showing that the presence of LSS, lumbar symptom, and deterioration of

QOL due to low back pain were associated with MCI and MMSE summary score. We showed that the LSS presence was significantly related to the MMSE summary score. Pain was a serious symptom secondary to neuropsychiatric symptoms in patients with dementia [30], and approximately 50% of people with dementia experienced pain regularly [31]. Chronic and neuropathic pain are particularly associated with cognitive impairment [6, 7]. A previous report showed that severe pain led to cognitive impairment in a longitudinal study [2]. In our study, no domain of VAS (low back pain, pain or numbness of lower limbs) was related with cognitive impairment; however, LSS presence was related to MMSE. The diagnosis tool for LSS used in this study consisted of 15 questions related to specific LSS symptoms, such as pain or numbness of buttock or lower limbs, and intermittent claudication. Thus, we suggested that LSS symptoms were related to cognitive impairment, not necessarily as single symptoms but as a combination of symptoms. Several reports have shown the relationship between physical dysfunction and cognitive impairment. Gait speed, standing balance, stand-up time, and leg strength declined more in cognitively impaired participants compared with healthy participants [1, 32, 33], and these physical functions deteriorated prior to cognitive decline [8]. In the current study, deterioration of QOL which was related to lumbar function, was associated with the prevalence of MCI and reduction in MMSE scores. The QOL decline related to lumbar function was evaluated by JOABPEQ, which is a self-administered questionnaire consisting of questions related to disorders of lumbar flexion, extension, and rotation due to low back pain. The disorder of lumbar motion leads to physical dysfunction, and these physical dysfunctions might cause cognitive impairment. There was a report that high-level physical activity reduced cognitive decline [34], while Sabia showed that exercise did not associate with cognitive decline in a longitudinal study [35]. It is controversial whether improving physical function makes cognitive impairment better. Persistent pain was associated with accelerated memory decline and increased probability of dementia [36]. However, whether the intervention of pain improves cognitive impairment remains unclear. Clarification of the relationship between cognitive impairment and lumbar degenerative disease is important for preventing the deterioration of QOL in a super-aging society. We should continue studying the relationship between lumbar degenerative disease and cognitive impairment longitudinally to clarify their causal relationship.

There were several limitations to our study. First, we used only the MMSE for evaluating cognitive function. Although MMSE is a standard measure of cognitive function, MMSE could evaluate only global cognitive function. Therefore, we could not study the relationships between lumbar degenerative disease and specific cognitive functions, such as memory and language, among others. Second, although more than 300 participants were included in this study, the study population may not be representative of the general population because participants were recruited from only one area of Japan. Third, we did not have detailed drug information about the participants. Analgesic drugs or other drugs, such as pregabalin, antidepressant, or benzodiazepines, would mask their symptoms, and we might have therefore underestimated cognitive impairment or lumbar degenerative disease. Fourth, we checked MMSE just one time in this study, but MMSE scores might be changed by mental and physical conditions. Further, we could not clarify the causal relationship between lumbar degenerative disease and cognitive impairment because this was a cross-sectional study. Hence, a longitudinal study investigating the causal relationship between lumbar degenerative disease and cognitive function is necessary in the future. Nevertheless, to the best of our knowledge, this is the first study to have investigated the relationship between lumbar degenerative disease and cognitive impairment in a Japanese population; our results provide valuable information to improve QOL of older adults in a super-aging society. This health promotion project started in 2005 and provided continuous data, enabling a longitudinal study to clarify further relationships between lumbar degenerative disease and cognitive impairment.

## Conclusions

Our results indicated that the comorbidity rate of MCI and LSS was 2.1%, the rate of MCI was 41% in participants with LSS, and the rate of LSS was 9.7% in participants with MCI in Japanese general population whose age was over 65 years. Deterioration of QOL related to lumbar function was associated with MCI. Deterioration of QOL related to lumbar function and LSS presence were associated with summary scores of MMSE. Clarification of the relationship between cognitive impairment and lumbar degenerative disease is important in preventing the deterioration of QOL in a super-aging society, and we should determine the causal relationship through a longitudinal study in the future. We recommend evaluating the cognitive function in patients with LSS, as the rate of MCI was over 40% of LSS in this study.

## Acknowledgments

The authors wish to thank Prof. Shigeyuki Nakaji and all staff and participants related to Iwaki Health Promotion Project in 2016.

## Author Contributions

**Conceptualization:** Kanichiro Wada, Toru Asari.

**Formal analysis:** Kazushige Koyama.

**Investigation:** Kazushige Koyama, Toru Asari.

**Methodology:** Gentaro Kumagai, Hitoshi Kudo, Sunao Tanaka, Toru Asari.

**Project administration:** Songee Jung, Masataka Ando.

**Supervision:** Yasuyuki Ishibashi.

**Writing – original draft:** Kazushige Koyama.

**Writing – review & editing:** Kanichiro Wada, Gentaro Kumagai, Hitoshi Kudo, Sunao Tanaka, Toru Asari, Songee Jung, Masataka Ando.

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
