## [Decision Letter · Decision Letter 0]

26 Jul 2021

PONE-D-21-11408

Associations between symptoms and quality of life in lumbar degenerative disease and cognitive dysfunction in a Japanese community: a cross-sectional study

PLOS ONE

Dear Dr. Wada,

Thank you for submitting your manuscript to PLOS ONE. After careful consideration, we feel that it has merit but does not fully meet PLOS ONE’s publication criteria as it currently stands. Therefore, we invite you to submit a revised version of the manuscript that addresses the points raised during the review process.

We look forward to receiving your revised manuscript.

Kind regards,

Masaki Mogi

Academic Editor

PLOS ONE

Additional Editor Comments (if provided):

#1. How about the use of analgesic drugs or other drugs such as pregabalin, anti-depressant, or benzodiazepines?

#2. Did the authors check MMSE just one time? Sometimes, MMSE score maybe changed.

#3. How about the depression scores in these subjects?

If there are not such data, it is better to describe them in Study Limitation.

Reviewers' comments:

Reviewer's Responses to Questions

**Comments to the Author**

1. Is the manuscript technically sound, and do the data support the conclusions?

Reviewer #1: Partly

2. Has the statistical analysis been performed appropriately and rigorously? 

Reviewer #1: Yes

3. Have the authors made all data underlying the findings in their manuscript fully available?

Reviewer #1: Yes

4. Is the manuscript presented in an intelligible fashion and written in standard English?

Reviewer #1: Yes

5. Review Comments to the Author

Reviewer #1: The paper is interesting but I have some suggestions that may help to improve its quality.

- The title needs to be rethought, as it is not entirely clear regarding the content of the paper .

- Regarding the introduction, it would be positive to include a brief explanation about the "super-aging society".

- In the introduction it is used the acronym ADL without its explanation.

- Regarding the diagnosis of LSS, which were the questions included in the self-administered questionnaire? It would be necessary to expand the information in this section.

- It would be easierfor the reader not using so many abbreviations. It is confusing and difficult to fully understand the paper in its current form.

- I would recommend to rewrite the Discussion section, as it repaets the main findings of the study but no proper discussion takes place.

6. PLOS authors have the option to publish the peer review history of their article (what does this mean?). If published, this will include your full peer review and any attached files.

Reviewer #1: **Yes: **Cristina Portellano-Ortiz

---

## [Author Response · Author response to Decision Letter 0]

16 Sep 2021

24th August, 2021

Masaki Mogi

Academic Editor

PLOS ONE

Dear Dr. Mogi:

We wish to re-submit the attached manuscript titled “Association between mild cognitive impairment and lumbar degenerative disease in a Japanese community: a cross-sectional study” as a Research Article. The manuscript ID is PONE-D-21-11408. We would like to change the title from “Associations between symptoms and quality of life in lumbar degenerative disease and cognitive dysfunction in a Japanese community: a cross-sectional study” to the above new title.

The manuscript has been rechecked and appropriate changes have been made in accordance with the reviewers’ suggestions. The responses to their comments have been prepared and are given below. For convenience, modified parts in the manuscript are in red-colored font.

We thank you and the reviewers for your thoughtful suggestions and insights, which have enriched the manuscript and produced a better and more balanced account of the research. We hope that the revised manuscript is now suitable for publication in your journal.

Sincerely,

Kanichiro Wada 

Department of Orthopedic Surgery, Hirosaki University Graduate School of Medicine

Zaifu-cho, Hirosaki, Aomori 036-8562, Japan

Department of Social Health, Hirosaki University Graduate School of Medicine

Zaifu-cho, Hirosaki, Aomori 036-8562, Japan

E-mail: wadak39@hirosaki-u.ac.jp

Tel: +81-172-39-5083

Fax: +81-36-3826

Editor and reviewer comments and replies

Comment 1 from editor

How about the use of analgesic drugs or other drugs such as pregabalin, anti-depressant, or benzodiazepines?

Reply 1 to editor

We thank the editor for providing advice on the current study, thereby improving the manuscript. The percentage of participants who used sleeping pills, analgesic drugs, and psychotropic drugs were 8.3% (28/336), 7.4% (25/336), and 1.5% (5/336). However, we did not have detailed drug information, such as the drug names, in this study. Analgesic drugs or other drugs would mask their symptoms, and we might underestimate cognitive impairment and/or lumbar degenerative disease. Thus, we have described the effect of the absence of drug information on the relationship between cognitive function and lumbar degenerative disease as a limitation of this study (page 22, lines 303-306).

Comment 2 from editor

Did the authors check MMSE just one time? Sometimes, MMSE score maybe changed.

Reply 2 to editor

Thank you for pointing out an important issue of our evaluations. We checked MMSE just one time in this study. As you mentioned, MMSE score may be changed by mental and physical conditions of individuals, and we have described this in the study limitations (page 22, line 306-307).

Comment 3 from editor

How about the depression scores in these subjects?

Reply 3 to editor

We appreciate the editor’s comment and agree that it is necessary to evaluate depression scores in this study. We have evaluated depression using The Center for Epidemiologic Studies Depression Scale and set a cut-off point at 16 points or greater. We have repeated the statistical analysis and revised the material and method section (page 8, lines 124-128), results, and tables (table 1 to 4).

Comment 4 from reviewer

The title needs to be rethought, as it is not entirely clear regarding the content of the paper.

Reply 4 to reviewer

Thank you for your valuable advice to improve the title of our study. We have changed the title to “Association between mild cognitive impairment and lumbar degenerative disease in a Japanese community: a cross-sectional study.”

Comment 5 from reviewer

Regarding the introduction, it would be positive to include a brief explanation about the "super-aging society".

Reply 5 to reviewer

Thank you for your valuable comment. “Super-aging society” means that the proportion of older adults whose age is >65 years old is over 21%. In Japan, the proportion of older adults is 26%. Accordingly, we have explained “super-aging society” in the introduction (page 3, lines 33-34).

Comment 6 from reviewer

In the introduction it is used the acronym ADL without its explanation.

Reply 6 to reviewer

We thank you for pointing out our insufficient description. We have spelled out ADL accordingly (activities of daily living) (page 3, line 48).

Comment 7 from reviewer

Regarding the diagnosis of LSS, which were the questions included in the self-administered questionnaire? It would be necessary to expand the information in this section.

Reply 7 to reviewer

Thank you for pointing out the lack of explanation about the questionnaire. We diagnosed LSS using a simple single sheet, self-administered, self-reported history questionnaire. It consisted of 15 questions associated with buttock and lower limb numbness or pain, and intermitted-claudication. Each question is scored from -1 to 5 points. We totaled the points of all questions and diagnosed LSS when the score was 13 points or greater. The sensitivity and specificity were 92.7% and 84.7% for diagnosing LSS in patients with lower limb symptoms due to LSS, arteriosclerosis obliterans, diabetes, or peripheral nerve diseases. These explanations have been added to the materials and methods (pages 5-6, lines 80-86).

Comment 8 from reviewer

It would be easier for the reader not using so many abbreviations. It is confusing and difficult to fully understand the paper in its current form.

Reply 8 to reviewer

We agree with your comment regarding not using too many abbreviations in this study. We used abbreviations; lumbar degenerative disease (LDD), lumbar spinal canal stenosis (LSS), mild cognitive impairment (MCI), quality of life (QOL), Japanese Orthopedic Association Back-Pain Evaluation Questionnaire (JOABPEQ), Mini Mental State Examination (MMSE), activities of daily living (ADL), visual analog scale (VAS), Kellgren-Lawrence grading (KL), diabetes (DM), and confidence interval (CI). We have reduced the number of abbreviations, other than the important key words in this study; LSS, MCI, QOL, JOABPEQ, MMSE, VAS and CI. We have reworded “LDD symptom” to “lumbar degenerative disease symptom.”

Comment 9 from reviewer

I would recommend to rewrite the Discussion section, as it repeats the main findings of the study but no proper discussion takes place.

Reply 9 to reviewer

We thank you for your valuable comment for improving the discussion of this study. We repeated the main findings of the study in the first paragraph, as well as in the middle of the second and third paragraphs and the conclusion; thus, we have deleted the first paragraph, which was the summary of the result. In the first paragraph, we discussed the prevalence of MCI and LSS combined. The comorbidity rate of MCI and LSS was not high (2.1%; however, the rate of MCI in participants with LSS　and the rate of LSS in participants with MCI were high (41%, 9.7%). According to a previous study, the risk factors of LSS presence, such as age or diabetes or low ankle-brachial index values, were similar to the risk factors of cognitive impairment. We have included this discussion in the first paragraph (pages 17-19, lines 220-257). In the second paragraph, we discussed that LSS presence and QOL decline due to low back pain were associated with MCI or MMSE summary score. It is clear that pain is related to cognitive impairment in the previous study. We suggested that a combination of LSS symptoms, such as pain or numbness of buttock or lower limbs or intermittent claudication, were related to cognitive impairment as with the previous study. It is clear that physical dysfunction associates with cognitive impairment. We suggest that physical dysfunction (limitation in range of lumbar motion) due to low back pain caused QOL decline and cognitive impairment. Several reports showed a causal relationship between pain or physical dysfunction and cognitive impairment; however, whether the intervention of pain or physical dysfunction improves cognitive impairment remains unclear. We have added that longitudinal study should be continued to clarify the causal relationship between lumbar degenerative disease and cognitive impairment (pages 19-21, lines 258-295).

---

## [Decision Letter · Decision Letter 1]

7 Oct 2021

Association between mild cognitive impairment and lumbar degenerative disease in a Japanese community: a cross-sectional study

PONE-D-21-11408R1

Dear Dr. Wada,

We’re pleased to inform you that your manuscript has been judged scientifically suitable for publication and will be formally accepted for publication once it meets all outstanding technical requirements.

Kind regards,

Masaki Mogi

Academic Editor

PLOS ONE

Additional Editor Comments (optional):

No further comment.

Reviewers' comments:

Reviewer's Responses to Questions

**Comments to the Author**

1. If the authors have adequately addressed your comments raised in a previous round of review and you feel that this manuscript is now acceptable for publication, you may indicate that here to bypass the “Comments to the Author” section, enter your conflict of interest statement in the “Confidential to Editor” section, and submit your "Accept" recommendation.

Reviewer #1: All comments have been addressed

2. Is the manuscript technically sound, and do the data support the conclusions?

Reviewer #1: Yes

3. Has the statistical analysis been performed appropriately and rigorously? 

Reviewer #1: Yes

4. Have the authors made all data underlying the findings in their manuscript fully available?

Reviewer #1: Yes

5. Is the manuscript presented in an intelligible fashion and written in standard English?

Reviewer #1: Yes

6. Review Comments to the Author

Reviewer #1: (No Response)

7. PLOS authors have the option to publish the peer review history of their article (what does this mean?). If published, this will include your full peer review and any attached files.

Reviewer #1: No

---

## [Editor Report · Acceptance letter]

11 Oct 2021

PONE-D-21-11408R1 

Association between mild cognitive impairment and lumbar degenerative disease in a Japanese community: a cross-sectional study 

Dear Dr. Wada:

I'm pleased to inform you that your manuscript has been deemed suitable for publication in PLOS ONE. Congratulations! Your manuscript is now with our production department. 

Kind regards, 

on behalf of

Dr. Masaki Mogi 

Academic Editor

PLOS ONE